# Adjuvant Sorafenib for Postoperative Patients with Hepatocellular Carcinoma and Macrovascular Invasion

Che-Jui Chang [1], Wei-Fan Hsu [1,2,3,*], Long-Bin Jeng [4], Hsueh-Chou Lai [1,3], Shih-Chao Hsu [4], Te-Hung Chen [4], Hung-Wei Wang [1,5] and Cheng-Yuan Peng [1,5]

[1] Center for Digestive Medicine, Department of Internal Medicine, China Medical University Hospital, Taichung 404327, Taiwan; 032600@tool.caaumed.org.tw (C.-J.C.)
[2] Graduate Institute of Biomedical Science, China Medical University, Taichung 404328, Taiwan
[3] School of Chinese Medicine, China Medical University, Taichung 404328, Taiwan
[4] Department of Surgery, China Medical University Hospital, Taichung 404327, Taiwan
[5] School of Medicine, China Medical University, Taichung 404328, Taiwan
* Correspondence: drizzt.hsu@gmail.com

**Abstract:** Hepatocellular carcinoma (HCC) is a leading cause of cancer-related mortality in Taiwan. Some patients with HCC are diagnosed with macrovascular invasion (MVI), which is associated with a poorer prognosis. In Taiwan, sorafenib is the first-line therapy for patients with advanced HCC. However, the efficacy of adjuvant sorafenib therapy remains unclear for the subset of patients with HCC and MVI who are eligible for surgery. Therefore, we investigated the potential benefit of adjuvant sorafenib therapy for patients with HCC and MVI after surgery. Our study showed that the lack of improved PFS or OS of adjuvant sorafenib challenged the therapeutic benefit of postoperative sorafenib. Alcohol consumption and an $\alpha$-fetoprotein level of $\geq 400$ ng/mL were independent predictors of overall survival (OS); however, adjuvant sorafenib therapy was not a predictor of progression-free survival (PFS) or OS. In conclusion, our study indicated that adjuvant sorafenib therapy did not provide PFS or OS benefits in patients with HCC and MVI.

**Keywords:** sorafenib; hepatocellular carcinoma; macrovascular invasion; post-operation

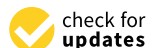



## 1. Introduction

Hepatocellular carcinoma (HCC) is a prominent contributor to cancer-related deaths globally and in Taiwan [1]. Surgical resection remains the primary curative approach for treating HCC.

However, some HCC patients are diagnosed with macrovascular invasion (MVI), and the overall survival (OS) in this subgroup of patients is grave [2]. Costentin et al. demonstrated that surgical resection did not provide additional survival benefits relative to sorafenib therapy in patients with HCC and MVI [3], and the therapeutic benefits of surgical resection in these patients remain inconclusive [2]. Therefore, the European Association for the Study of the Liver (EASL) suggested against surgical resection as a standard of practice in this subgroup of patients [4]. Although the American Association for the Study of Liver Disease (AASLD) did not directly suggest against surgical resection for HCC patients with MVI, extended indications for surgical resection in this subgroup of patients should be made in experienced centers with multidisciplinary support [5].

Sorafenib was the first-line therapy for patients with advanced HCC [6,7] until the introduction of atezolizumab plus bevacizumab [8]. Sorafenib is a safe and effective drug for patients who underwent surgical resection [9,10]. However, early tumor recurrence after hepatectomy is a major problem for patients with HCC and MVI. In a randomized phase 3 placebo-controlled and double-blind trial, adjuvant sorafenib for HCC after resection or ablation (STORM, $n$ = 1114) showed that sorafenib had no benefit in the adjuvant setting on progression-free survival (PFS) and OS [11]. However, the trial enrolled most patients

at low risk of HCC recurrence (only 32–33% of patients with microvascular invasion). Therefore, a Chinese consensus also recommended against adjuvant molecular targeted therapies in HCC patients with a low risk of recurrence after radical resection [12]. A small pilot study (*n* = 31) in Taiwan focused on patients with risks of HCC recurrence, such as poor differentiation, microvascular invasion, and satellite nodules, revealed a significant impact of adjuvant sorafenib on recurrence rate and PFS [13]. Therefore, the efficacy of adjuvant sorafenib is a topic worth exploring in the subset of patients with HCC and MVI who are eligible for surgery (e.g., unilateral HCC with good performance status).

In the present study, we retrospectively collected the data of patients diagnosed with HCC and MVI who underwent survival resection with or without adjuvant therapy to investigate the potential benefits of adjuvant sorafenib.

## 2. Materials and Methods

### 2.1. Patients

Our retrospective analysis included 43 postoperative patients with major portal or hepatic venous tumor thrombosis from March 2013 to March 2022. Patients were excluded if they were aged < 18 years, had an immunodeficiency virus infection, or had malignancies other than HCC.

We collected baseline data on demographic characteristics, including whole blood count, biochemical data, coagulation, α-fetoprotein (AFP), hepatitis B virus (HBV) infection, hepatitis C virus (HCV) infection, alcohol used, status of diabetes mellitus (DM), and liver cirrhosis (LC). The present study received approval from the Research Ethics Committee of China Medical University Hospital (CMUH) and was conducted in adherence to the principles outlined in the 1975 Declaration of Helsinki (CMUH108-REC3-140). The need for informed consent was waived as the personal information of all included patients was anonymized through encryption.

### 2.2. Diagnosis and Treatment of HCC

The study hospital's central laboratory analyzed blood biochemistry data (obtained using a Beckman Coulter analyzer) and whole blood count measurements (obtained using the Sysmex HST series). Alcohol consumption was counted on a daily basis; exceeding 30 g for males and 20 g for females were categorized as alcohol consumers [14]. The presence of hepatitis B surface antigen in the serum was used to define HBV infection and the presence of anti-HCV antibody plus detectable HCV RNA in the serum was used to define HCV infection (Roche Diagnostics). Albumin–bilirubin scores and grades were calculated per the protocol of another study [15]. The identification of HCC relied on histological observations or the characteristic radiological manifestations confirmed via a minimum of two imaging studies, including contrast-enhanced dynamic computed tomography (CT), magnetic resonance imaging (MRI), hepatic angiography, and abdominal ultrasound [4,16]. The assessment of MVI severity was based on contrast-enhanced dynamic CT, MRI, or postoperative histologic findings. Each grade is defined as follows: Vp1, subsegmental or presence of a tumor thrombus distal to the second-order branches of the portal vein; Vp2, segmental or presence of a tumor thrombus in the second-order branches of the portal vein; Vp3, presence of a tumor thrombus in the first-order branches of the portal vein; and Vp4, presence of a tumor thrombus in the main trunk or contralateral branch [17]. The determination of LC status relied on unequivocal clinical, ultrasonographic, and histological evidence.

The multidisciplinary HCC team at CMUH provided therapeutic recommendations for HCC patients based on the guidelines of the AASLD and EASL [4,5]. The proficient surgeons at CMUH assessed the necessary extent of surgical resection, while the prescription of sorafenib dosage followed recommended protocols [18]. Patients who underwent adjuvant sorafenib therapy after surgical resection were assigned to an adjuvant sorafenib group and those who only underwent surgical resection were assigned to a control group.

*2.3. Statistical Analysis*

Continuous variables are reported as medians (first to third quartiles), and categorical variables are presented as frequencies (percentages). Data were censored until death, loss, or the conclusion of the follow-up period (i.e., until 31 March 2023), whichever event occurred at first. Mann–Whitney U tests were conducted to evaluate the differences in continuous variables between groups. The variables with *p*-values of <0.20 determined via univariate analysis were then included in multivariable Cox regression models to investigate their associations with PFS or OS. Statistical analyses were performed using SPSS (version 25.0, IBM Corp., Armonk, NY, USA). A two-sided *p*-value less than 0.05 was considered statistically significant.

## 3. Results

Of the 43 patients included in the present study, 31 (72.1%) were men, and their median age was 58 (52–65) years. Among the patients, 10 (23.3%) were alcohol consumers; 27 (62.8%) and 11 (26.2%) had HBV and HCV infection, respectively; and 18 (41.9%) had liver cirrhosis. Thirty-five (83.3%) and seven (16.7%) patients had albumin–bilirubin grades of 1 and 2, respectively. The median AFP level and maximal tumor size were 98.96 (10.93–2786.89) ng/mL and 8.0 (4.9–11.5) cm, respectively, and the patients were all classified as having Barcelona Clinic Liver Cancer stage C HCC. Among the patients, 4 (9.3%) had Vp2 status, 16 (37.2%) had Vp3 status, 6 (14.0%) had Vp4 status, and 17 (39.5%) had hepatic venous thrombosis. PFS and OS were 10.00 (6.04–13.96) months and 19.77 (1.71–79.23) months, respectively.

The adjuvant sorafenib group and control group comprised 10 and 33 patients, respectively. Those who underwent adjuvant sorafenib therapy were older adults, with a low proportion of individuals with HBV infection. The PFS and OS of the two groups were not significantly different (Table 1).

**Table 1.** Demographics, baseline characteristics, and therapeutic response of patients.

| Character | All (*n* = 43) | Adjuvant Sorafenib (*n* = 10) | Control (*n* = 33) | *p*-Value |
|---|---|---|---|---|
| Age (years) | 58 (52–65) | 66 (62–72) | 57 (50–63) | 0.007 |
| Sex (male), *n* (%) | 31 (72.1) | 5 (50) | 26 (78.8) | 0.175 |
| Body mass index (kg/m$^2$) | 23.57 (21.28–26.39) | 25.44 (21.50–27.05) | 23.46 (21.06–25.90) | 0.232 |
| Platelet count ($10^9$/L) | 198 (162–253) | 210 (148–399) | 197 (166–249) | 0.380 |
| AST (U/L) | 41 (29–66) | 34 (28–74) | 41 (32–62) | 0.645 |
| ALT (U/L) | 35 (24–59) | 29 (21–41) | 38 (28–63) | 0.079 |
| Total bilirubin (mg/dL) | 0.80 (0.6–1.00) | 0.76 (0.45–1.27) | 0.8 (0.62–0.98) | 0.729 |
| Albumin (g/dL) | 4.3 (4.1–4.6) | 4.2 (3.7–4.4) | 4.4 (4.1–4.6) | 0.074 |
| Creatinine (mg/dL) | 0.89 (0.79–0.98) | 0.89 (0.74–0.98) | 0.89 (0.81–0.99) | 0.774 |
| Etiology | | | | |
| Alcohol | 10 (23.3) | 1 (10) | 9 (27.3) | 0.419 |
| HBV | 27 (62.8) | 3 (30) | 24 (72.7) | 0.041 |
| HCV | 11 (26.2) | 4 (40) | 7 (21.9) | 0.406 |
| Diabetes mellitus | 12 (27.9) | 1 (10) | 11 (33.3) | 0.273 |
| Liver cirrhosis | 18 (41.9) | 6 (60) | 12 (36.4) | 0.273 |
| ALBI grade | | | | 0.154 |
| 1 | 35 (83.3) | 6 (60) | 29 (90.6) | |
| 2 | 7 (16.7) | 4 (40) | 3 (9.4) | |

**Table 1.** *Cont.*

| Character | All (*n* = 43) | Adjuvant Sorafenib (*n* = 10) | Control (*n* = 33) | *p*-Value |
|---|---|---|---|---|
| AFP (ng/mL) | 98.96 (10.93–2786.89) | 24.02 (8.74–545.98) | 249.88 (11.15–5591.50) | 0.202 |
| AFP > 400 ng/mL | 15 (35.7) | 2 (20) | 13 (40.6) | 0.343 |
| PFS (months) * | 10.00 (6.04–13.96) | 14.10 (5.74–22.46) | 10.00 (4.14–15.86) | 0.268 |
| OS (months) * | 19.77 (1.71–79.23) | 19.95 (0.00–69.57) | 20.16 (0.96–79.98) | 0.204 |

Data presented as median (first quartile–third quartile). * Data presented as median (95% confidence interval). Abbreviation: AFP, α-fetoprotein; ALBI, albumin–bilirubin; ALT, alanine aminotransferase; AST, aspartate aminotransferase; HBV, hepatitis B virus; and HCV, hepatitis C virus.

Our univariate Cox regression analysis did not reveal a significant association between PFS and adjuvant sorafenib therapy (hazard ratio (HR), 0.733; 95% confidence interval (CI), 0.291–2.052; *p* = 0.606; Table 2). In the other analysis, alcohol consumption and an AFP level of ≥400 ng/mL were associated factors of OS in the univariate Cox regression analysis (*p* < 0.200). Through multivariable Cox regression analysis, we identified alcohol consumption (HR, 2.930; 95% CI, 1.150–7.467; *p* = 0.024) and AFP ≥ 400 ng/mL (HR, 2.614; 95% CI, 1.033–6.614; *p* = 0.043) as significant independent predictors of OS (Table 3). By contrast, adjuvant sorafenib therapy was not a significant predictor of PFS or OS.

**Table 2.** Factors associated with progression-free survival.

| Characteristic | | Univariate Analysis | | Multivariate Analysis | |
|---|---|---|---|---|---|
| | | HR (95% CI) | *p* Value | HR (95% CI) | *p* Value |
| Age (year) | | 1.005 (0.974–1.037) | 0.757 | | |
| Sex | M vs. F | 1.066 (0.476–2.387) | 0.877 | | |
| Alcohol | Yes vs. no | 1.240 (0.578–2.660) | 0.580 | | |
| HBV | Yes vs. no | 1.119 (0.522–2.400) | 0.773 | | |
| HCV | Yes vs. no | 1.583 (0.709–3.537) | 0.263 | | |
| DM | Yes vs. no | 0.936 (0.432–2.027) | 0.866 | | |
| AFP (ng/mL) | <400 vs. ≥400 | 1.210 (0.564–2.599) | 0.625 | | |
| AST (U/L) | >40 vs. ≤40 | 0.529 (0.242–1.158) | 0.111 | | |
| ALT (U/L) | >40 vs. ≤40 | 0.799 (0.382–1.669) | 0.550 | | |
| ALBI grade | 2 vs. 1 | 0.602 (0.208–1.739) | 0.349 | | |
| Adjuvant sorafenib | Yes vs. no | 0.773 (0.291–2.052) | 0.606 | | |

Abbreviations: AFP, α-fetoprotein; ALBI, albumin–bilirubin; ALT, alanine aminotransferase; AST, aspartate aminotransferase; DM, diabetes mellitus; HBV, hepatitis B virus; HCV, hepatitis C virus; and M vs. F, male vs. female.

**Table 3.** Factors associated with overall survival.

| Characteristic | | Univariate Analysis | | Multivariate Analysis | |
|---|---|---|---|---|---|
| | | HR (95% CI) | *p* Value | HR (95% CI) | *p* Value |
| Age (year) | | 0.977 (0.942–1.013) | 0.202 | | |
| Sex | M vs. F | 2.025 (0.675–6.077) | 0.208 | | |
| Alcohol | Yes vs. no | 2.351 (0.956–5.783) | 0.063 | 2.930 (1.150–7.467) | 0.024 |
| HBV | Yes vs. no | 1.571 (0.566–4.357) | 0.386 | | |
| HCV | Yes vs. no | 0.710 (0.233–2.167) | 0.548 | | |
| DM | Yes vs. no | 1.221 (0.494–3.019) | 0.665 | | |

**Table 3.** *Cont.*

| Characteristic | | Univariate Analysis | | Multivariate Analysis | |
|---|---|---|---|---|---|
| | | HR (95% CI) | *p* Value | HR (95% CI) | *p* Value |
| AFP (ng/mL) | <400 vs. ≥400 | 2.244 (0.903–5.575) | 0.082 | 2.614 (1.033–6.614) | 0.043 |
| AST (U/L) | >40 vs. ≤40 | 1.234 (0.485–3.139) | 0.659 | | |
| ALT (U/L) | >40 vs. ≤40 | 0.903 (0.368–2.212) | 0.823 | | |
| ALBI grade | 2 vs 1 | 1.442 (0.475–4.375) | 0.518 | | |
| Adjuvant sorafenib | Yes vs no | 0.710 (0.160–3.155) | 0.653 | | |

Abbreviations: AFP, α-fetoprotein; ALBI, albumin–bilirubin; ALT, alanine aminotransferase; AST, aspartate amino-transferase; DM, diabetes mellitus; HBV, hepatitis B virus; HCV, hepatitis C virus; and M vs. F, male vs. female.

The adjuvant sorafenib and control groups did not exhibit significant differences in PFS (adjuvant sorafenib vs. control, 14.10 (5.74–22.46) months vs. 10.00 (4.14–15.86) months; *p* = 0.268; Figure 1A) and OS (adjuvant sorafenib vs. control, 19.95 (0.00–69.57) months vs. 20.16 (0.96–79.98) months; *p* = 0.204; Figure 1B).

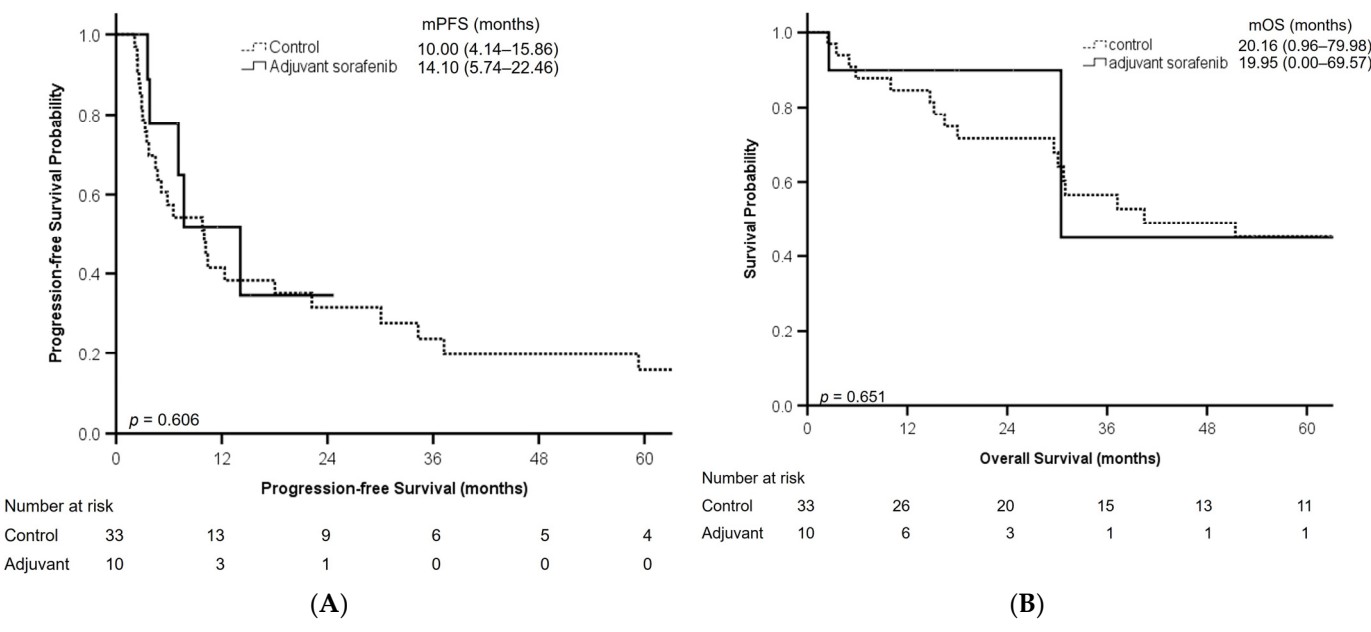

**Figure 1.** Progression-free survival (**A**) and overall survival (**B**) of patients in adjuvant sorafenib and control groups. Abbreviations: mOS, median overall survival; mPFS, median progression-free survival.

## 4. Discussion

HCC is a leading cause of cancer-related death in Taiwan because of the endemic prevalence of HBV infection. In response, the Taiwanese government, particularly the National Health Insurance Administration, has committed considerable resources to improve public health challenges, including a program for universal hepatitis B vaccination (introduced in 1984), nationwide National Health Insurance (introduced in 1995), and a national program for viral hepatitis treatment (introduced in 2004) [19]. However, some patients do not take advantage of Taiwan's universal health care services and develop advanced HCC that could have been prevented. In the present retrospective study, we discovered that most patients exhibited risk factors associated with HCC including HBV infection, HCV infection, and alcohol consumption. Moreover, alcohol consumption has been identified as an independent risk factor for OS in patients with HCC and MVI.

According to the current guidelines, including the guidelines of the EASL [4], the Barcelona Clinic Liver Cancer framework [20], and the AASLD [5], for individuals with HCC and MVI, the recommended therapy is to opt for systemic therapy instead of surgical resection. However, patients in the previous study who underwent sorafenib monotherapy achieved a median overall survival (OS) ranging from 6.5 to 10.7 months [9,10]. The AASLD and EASL mentioned that surgical resection could only be considered for HCC patients with Vp 1 or 2 MVI in high-volume centers or research settings [4,5]. This finding suggests that alternative interventions should be developed for this subgroup of patients. Several clinical trials investigating neoadjuvant or adjuvant immunotherapy for resectable HCC are still undergoing, such as NCT05389527 (neoadjuvant pembrolizumab plus lenvatinib), NCT04658147 (neoadjuvant or adjuvant nivolumab or nivolumab plus relatlimab), and NCT04615143 (neoadjuvant tislelizumab or tislelizumab plus lenvatinib). We also discovered that adjuvant sorafenib therapy did not provide significantly superior survival benefits relative to surgical resection only (adjuvant sorafenib vs. surgical resection only, 19.95 (0.00–69.57) vs. 20.16 (0.96–79.98) months, $p = 0.204$), and the median OS of the study patients is comparable to those reported in other studies [8,21,22]. This result may be related to the limited number of patients in the present study.

In recent years, several studies have indicated that patients with HCC and MVI receiving standard sorafenib treatment may experience improved OS. Therefore, the National Comprehensive Cancer Network guidelines have incorporated sorafenib as a treatment option for these patients, with numerous ongoing clinical trials exploring related aspects [8,21,22]. Costentin et al. performed a propensity score analysis to compare the clinical benefit between sorafenib and surgical resection in patients with HCC and MVI ($n = 143$) [3], and the patients who received surgical resection had no longer median OS than those who received sorafenib (12.0 months vs 9.7 months, $p = 0.682$). Chen et al. reported that adjuvant lenvatinib in combination with transarterial chemoembolization (TACE) had longer disease-free survival than adjuvant TACE alone in patients with high postoperative recurrence risk ($n = 184$) (NCT03838796). Our study did not show the therapeutic benefit of adjuvant sorafenib in postoperative patients with HCC and MVI. This could be associated with the patient's clinical condition and inherent limitations of our research (discussed in the following paragraph). As mentioned above, the AASLD and EASL only considered surgical resection for HCC patients with Vp1–2 MVI in high-volume centers or research settings. However, some guidelines from China [12], Hong Kong [23], and Japan [24] have alternative opinions. These guidelines indicated that surgical resection may be considered in some selected patients with HCC and MVI to prolong the OS and possible life quality. Our study also enrolled 37.2% ($n = 16$) and 14% ($n = 6$) of patients with Vp3 and Vp4 MVI, respectively.

There were some limitations in this study. First, the limited sample size constrained our capacity to identify statistically significant differences relating to key variables and to perform detailed comparisons. Second, the follow-up period was short. Thus, future studies should implement longer follow-up periods. Third, we did not examine the implementation of systemic therapy following sorafenib therapy. However, given that adjuvant sorafenib therapy did not provide additional therapeutic benefits, the benefits of subsequent systemic therapy are likely to be limited.

## 5. Conclusions

We revealed that for patients with HCC and MVI, adjuvant sorafenib therapy does not provide benefits with respect to PFS and OS. Nevertheless, further studies with larger sample sizes and longer follow-up periods are warranted.

**Author Contributions:** C.-J.C. collected the data and wrote the paper. W.-F.H. designed the study, analyzed the data, performed the research, supervised the study, and critically revised the manuscript. L.-B.J., H.-C.L., S.-C.H., T.-H.C., H.-W.W. and C.-Y.P. had substantial contributions to the conception, acquisition of data for the work, and revising the intellectual content. All authors have read and agreed to the published version of the manuscript.

**Funding:** This study was supported by a grant (DMR-113-013) from China Medical University Hospital, Taichung, Taiwan.

**Institutional Review Board Statement:** This study was conducted in accordance with the 1975 Declaration of Helsinki and was approved by the Research Ethics Committee of CMUH, Taichung, Taiwan (CMUH108-REC3-140).

**Informed Consent Statement:** The need for informed consent was waived because personal information was encrypted for privacy protection.

**Data Availability Statement:** The data presented in this study are available upon request from the corresponding author.

**Conflicts of Interest:** The authors declare no conflict of interest.

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
