# Peer review of "Adjuvant Sorafenib for Postoperative Patients with Hepatocellular Carcinoma and Macrovascular Invasion"

_curroncol, doi:10.3390/curroncol30120737_

Round 1
Reviewer 1 Report
Comments and Suggestions for Authors
1). This work is not novel. The Asia Pacific Phase III Sorafenib trial analysed the effect of sorafenib on patients with advanced HCC, including post resection patients and found a benefit, contrary to the study reported here. This needs to be added to and discussed in the Discussion section.
2). The Introduction and Discussion sections need to be worked on further. There are conflicting reports on the effects of adjuvant sorafenib treatment on overall survival in HCC with microvascular invasion. This should also be discussed.
3). As the authors themselves point out, the sample size is inadequate, especially taking into consideration other background characteristics that may compound their observations.
Comments on the Quality of English LanguageThe overall writing style needs some improvement to make it less conversational, yet more descriptive while keeping it succinct at the same time.
Author Response
Dear Section Editor and Reviewers,
We highly appreciate your constructive comments and valuable suggestions which have been incorporated into the revised manuscript. The following are our point-by-point responses to your comments. We look forward to hearing from you soon.
Sincerely yours,
Wei-Fan Hsu, MD, MMS
[reviewer 1]
1). This work is not novel. The Asia Pacific Phase III Sorafenib trial analysed the effect of sorafenib on patients with advanced HCC, including post-resection patients and found a benefit, contrary to the study reported here. This needs to be added to and discussed in the Discussion section.
2). The Introduction and Discussion sections need to be worked on further. There are conflicting reports on the effects of adjuvant sorafenib treatment on overall survival in HCC with microvascular invasion. This should also be discussed.
Response: Thank you for your critical comments. We reviewed the previous study, including the Asia-Pacific Phase III Sorafenib trial (Lancet Oncol 2009;10:25). The study did enroll patients with advanced HCC, including those who had received previous local therapy, such as surgery, radiotherapy, et al. However, we did not identify subgroup analysis and SHARP-related articles focusing on adjuvant sorafenib. We added content about surgery for patients with HCC and MVI from the AASLD and EASL and about adjuvant sorafenib from the trial of STORM, some pilot studies, and clinical trials in the Introduction and Discussion sections.
3). As the authors themselves point out, the sample size is inadequate, especially taking into consideration other background characteristics that may compound their observations.
Response: Thank you for your vital comment. Our study had several limitations, and we pointed out in the Discussion section (lines 207–209, page 5). However, our findings prompt essential considerations for alternative therapies or factors influencing postoperative outcomes in patients with HCC and MVI. Further studies are warranted.

Reviewer 2 Report
Comments and Suggestions for Authors
The study entitled “Adjuvant sorafenib for postoperative patients with hepatocellular carcinoma and macrovascular invasion” reports the investigation of efficacy of adjuvant sorafenib for HCC patients with MVI.
I find the article excellent. No major issues were detected.
One thing needs to be mentioned: To my understanding, the article category selected for this manuscript is not the right one. The given manuscript much rather is a communication article than a commentary because the authors present a retrospective research study which is not typical for non-research commentaries.
Minor issues:
1. Page 3, line 4: For understanding: Do the terms “VP2”, “VP3” and “VP4” mean viral capsid proteins?
2. Page 3, Table 1: Please add “AST, aspartate aminotransferase” to caption below the table 1.
3. Page 5, line 3: Spelling error: “sprogression”. Please correct.
4. Page 5, line: Syntax error: “This finding suggests that alternative interventions should be developed for this subgroup of patients is crucial.” Remove “is crucial”.
Author Response
Dear Section Editor and Reviewers,
We highly appreciate your constructive comments and valuable suggestions which have been incorporated into the revised manuscript. The following are our point-by-point responses to your comments. We look forward to hearing from you soon.
Sincerely yours,
Wei-Fan Hsu, MD, MMS
[reviewer 2]
I find the article excellent. No major issues were detected.
One thing needs to be mentioned: To my understanding, the article category selected for this manuscript is not the right one. The given manuscript much rather is a communication article than a commentary because the authors present a retrospective research study, which is not typical for non-research commentaries.
Response: Thank you very much for your favorable comment. We modified the manuscript to a communication article (line 1, page 1).
[Minor issues]
- Page 3, line 4: For understanding: Do the terms “VP2”, “VP3” and “VP4” mean viral capsid proteins?
Response: Thank you for your critical comment. We mentioned the definition of the Vp0–4 in the Material and Method section (lines 88–92, page 2).
- Page 3, Table 1: Please add “AST, aspartate aminotransferase” to caption below the table 1.
- Page 5, line 3: Spelling error: “sprogression”. Please correct.
- Page 5, line: Syntax error: “This finding suggests that alternative interventions should be developed for this subgroup of patients is crucial.” Remove “is crucial”.
Response: Thank you for your kind corrections. We have modified these points.

Reviewer 3 Report
Comments and Suggestions for Authors
The study exploring adjuvant sorafenib therapy in HCC patients with MVI offers crucial insights. While it successfully identifies predictors of overall survival—alcohol consumption and α-fetoprotein levels—the lack of improved PFS or OS challenges the therapeutic benefit of sorafenib post-surgery. The study's limitations, including sample size and patient diversity, merit attention. However, these findings prompt essential considerations for alternative therapies or factors influencing HCC outcomes post-surgery in patients with MVI, warranting further investigation.
Author Response
Dear Section Editor and Reviewers,
We highly appreciate your constructive comments and valuable suggestions which have been incorporated into the revised manuscript. The following are our point-by-point responses to your comments. We look forward to hearing from you soon.
Sincerely yours,
Wei-Fan Hsu, MD, MMS
[Reviewer 3]
The study exploring adjuvant sorafenib therapy in HCC patients with MVI offers crucial insights. While it successfully identifies predictors of overall survival—alcohol consumption and α-fetoprotein levels—the lack of improved PFS or OS challenges the therapeutic benefit of sorafenib post-surgery. The study's limitations, including sample size and patient diversity, merit attention. However, these findings prompt essential considerations for alternative therapies or factors influencing HCC outcomes post-surgery in patients with MVI, warranting further investigation.
Response: Thank you for your honest and critical comment. We mentioned the study's limitations, including the small sample size, in the Discussion section (Lines 166–167, page 5). We also added content about surgery for patients with HCC and MVI from the AASLD and EASL and about adjuvant sorafenib from the trial of STORM, some pilot studies, and clinical trials in the Introduction and Discussion sections.

Round 2
Reviewer 1 Report
Comments and Suggestions for Authors
The authors have addressed my previous comments. Although the concerns such as the low sample size in their study still remain, the study may be published in its revised form.